# Gray-Scale, Color Doppler, Spectral Doppler, and Contrast-Enhanced Renal Artery Ultrasound: Imaging Techniques and Features

**DOI:** 10.3390/jcm11143961

**Published:** 2022-07-07

**Authors:** Byung Kwan Park

**Affiliations:** Department of Radiology, Samsung Medical Center, Sungkyunkwan University School of Medicine, Seoul 06351, Korea; rapark@skku.edu; Tel.: +82-2-3410-6457

**Keywords:** renal artery stenosis, kidney, gray-scale ultrasound, doppler ultrasound, contrast-enhanced ultrasound

## Abstract

Renal artery stenosis (RAS) is increasingly being detected in elderly patients as life expectancy increases. RAS induces hypertension or reduces renal function. Computed tomography or magnetic resonance angiography are objective in detecting RAS but may cause iodine-induced nephrotoxicity or nephrogenic systemic fibrosis in patients with RAS. Ultrasound (US) is, by contrast, a noninvasive and real-time imaging modality useful in patients with reduced renal function. Renal US is not as sensitive for detecting RAS because this technique indirectly assesses the renal artery by analyzing intrarenal hemodynamic changes. Although, ideally, US would be used to directly evaluate the renal artery, its current utility for RAS detection remains unclear. The purpose of this review is to introduce how to assess renal artery with US, to describe imaging features of renal artery US, to compare renal artery US and renal US, and to show how to perform work-up in patients in whom RAS is suspected.

## 1. Introduction

Renal artery stenosis (RAS) is one of the major causes of secondary hypertension and renal impairment [1,2]. There are many imaging studies on detecting RAS with ultrasound (US) [3,4,5], computed tomography (CT) [6,7], magnetic resonance imaging (MRI) [6,7,8], digital subtraction angiography (DSA) [9,10,11], and angiotensin-converting enzyme inhibitor scintigraphy [12,13]. CT or MRI is preferred because radiologists are familiar with CT and MR angiography. However, these examinations require the use of intravascular contrast material for evaluating the diameter of the renal artery. Given that these patients frequently have decreased renal function, serious complications can be induced by the intravascular administration of iodine [14,15,16] or gadolinium contrast material [8,17,18].

US is a noninvasive, real-time examination method for detecting RAS. Recently available US scanners enable the depiction of small vessels or organs. Gray-scale US can assess the morphology of the renal artery and kidney. Hemodynamic changes in the renal artery and kidney are evaluated with color and spectral Doppler US. Contrast-enhanced US may directly show the diameter change in the renal artery with intravascular contrast material that is not harmful to patients with poor renal function. Therefore, US is a useful examination method for detecting RAS, regardless of patient renal function. Previously published papers showed the utility of intrarenal Doppler US for detecting RAS because of their poorly depicted renal artery [3,5,19,20,21]. These investigations used outdated US scanners, which had many limitations when directly depicting the renal artery. Recent US scanners have provided higher image resolution or tissue contrast than before. Therefore, the renal artery can be directly assessed with US scanners, but the role of renal artery US in assessing RAS remains unclear. There are many investigations dealing with direct assessment of the renal artery with US [4,22,23,24,25,26,27,28,29,30,31]. However, they mainly focused on the imaging features of Doppler US alone, and did not describe the imaging features of gray-scale US and contrast-enhanced US. In addition, technical tips for renal artery US are seldom seen because the frequency shifts are often too difficult to directly detect from the renal artery, which is severely stenotic. They did not clearly compare the renal artery and intrarenal assessments in terms of US techniques and clinical outcomes. Other limitations include the operator dependence, necessary education, and learning curve for RAS scanning. The recent US scanners provide high-frame rate resolution when measuring various perfusion parameters for RAS scanning [32].

The purpose of this review is to define how to assess the renal artery with gray-scale, color Doppler, spectral Doppler, or contrast-enhanced US, to describe imaging features of the renal artery US, to compare renal artery US, and renal US, and to show how to perform work-up in patients in whom RAS is suspected.

## 2. Clinical Aspects of RAS

There have been no population-based studies providing accurate data regarding the prevalence of renal artery stenosis [33,34,35]. Among the Medicare population, the prevalence of clinically manifested RAS is reported as 0.5% overall and 5.5% among patients with chronic kidney disease [36]. Among 2167 consecutive autopsy patients, artherosclerotic RAS was found in 14.7%, 28.6%, and 23.9% of patients with hypertension, renal insufficiency, and aortic aneurysm, respectively [33]. The most common etiology of RAS is atherosclerosis commonly involving the proximal one-third of the renal artery [1,35,36,37]. Fibromuscular dysplasia is one etiology for RAS, but significantly differs from atherosclerosis in terms of patient demographics, RAS location, imaging features, and intervention procedures (Table 1) [38,39,40]. In case of stenosis from fibromuscular dysplasia, RAS may involve the intermediate and distal parts of the renal artery, making diagnosis with US and color Doppler more difficult. The incidence of RAS is increasing because of longer life expectancies. Patients with RAS have clinical manifestations including abrupt aggravation of hypertension or renal function despite appropriate medication, and extensive arterial occlusive disease. Most of all, radiologic examinations should be performed to detect RAS in patients with these clinical features if they have unilateral small kidney [41,42,43].

## 3. Renal Artery US vs. Renal US

Renal artery US is a more-skilled technique than renal US because the renal artery is a small vessel that is deep-seated in the retroperitoneal space (Table 2). For this reason, many radiologists or sonographers rely on renal US to identify RAS. Understanding the anatomical characteristics and differences between the right and left renal arteries is essential for properly conducting renal artery US. Breath-holding is not necessary during renal artery US (Table 2). Accordingly, renal artery US appears to be better for detecting RAS in patients who cannot easily control respiration on their own. However, this US examination has several limitations: First, even though the main renal artery can be assessed with renal artery US, the assessment of segmental or subsegmental renal arteries is limited. Second, assessing renal arteries can be technically difficult because of poor sonic windows that result from bowel gas, poor image resolution, or weak frequent shift. Left RAS is more difficult to detect than right RAS. The left renal artery is farther from the transducer, more frequently obscured by bowel loops, and travels straighter without angulation. Third, multiple renal arteries are harder to detect with renal artery US because each is smaller than a single renal artery. Finally, renal artery US is more influenced by patient body mass index than renal US.

Renal US requires less difficult techniques than renal artery US. However, breath-holding is essential to acquire an optimal Doppler spectrum which is important when calculating acceleration time, rate, and resistive index. This quantitative measurement is a key to precisely identify delayed and weak pulse in patients with RAS. Therefore, renal US has a limitation in patients who have shortness of breath or respiratory distress.

## 4. Renal Artery US: Imaging Techniques

Renal artery US is not established terminology on PubMed; there are no papers defining it, even though many investigations have demonstrated the utility of US in assessing the velocity of bilateral renal arteries. It can be defined as an US technique used to directly assess the renal artery. The renal artery is not easy to detect with US because it is a deeply situated small vessel [23]. The right and left renal arteries are sited posterior to the left renal vein. Therefore, to assess RAS, the first step is to find the left renal vein [44]. The right renal artery arises from 9–12 o’clock of the aorta and passes behind the inferior vena cava (Figure 1). These anatomical characteristics result in the focal angulation of the right renal artery, in which blood flow is clearly visible because of the good frequency shift. In contrast, the left renal artery arises from 2–5 o’clock of the aorta and travels away from the transducer (Figure 1). For these reasons, the frequency shifts of the left renal artery are weaker than those of the right renal artery, so the left RAS is more difficult to detect than the right RAS with gray-scale, Doppler, and contrast-enhanced US. However, US has difficulties in detecting RAS in the aberrant renal artery, accessary renal artery, and polar branches.

## 5. Renal Artery US: Imaging Features

### 5.1. Gray-Scale US

Gray-scale US is not well-known to be particularly useful in detecting RAS. Previously published studies did not clearly describe the imaging features of gray-scale US, but rather those of Doppler US. Renal artery occlusion was not directly assessed even in an animal study [45]. However, current gray-scale US has the potential to assess renal arteries directly because the ongoing development of US scanners is providing higher resolution imaging than before (Table 3). Transabdominal US can be used to evaluate proximal and middle segments of renal arteries. Transrenal US can be used to assess the distal segment of renal arteries in the flank. The location, number, and length can be shown depending on the patient’s obesity or bowel gas (Figure 2). Using a renal artery stent does not accurately detect RAS when it is invisible due to posterior sonic shadowing. Significant RAS is almost always accompanied by poststenotic dilatation [40]. High-speed and high-pressure blood flow, created by passing through the RAS, may contribute to expanding the lumen of the renal artery by means of repeatedly beating the endothelium. Calcified plaque, which is an atherosclerosis resulting from endothelial damage, is frequently encountered in RAS [46,47].

Gray-scale US can show indirect signs of RAS in the kidney (Table 3). The size of an RAS-involved kidney is smaller than the other uninvolved kidney [42] (Figure 2). The renal cortex becomes thinner than the medulla because the former is more susceptible to hypoxic damage than the latter. The hypoxic cortex becomes hyperechoic, so that the cortico-medullary differentiation of an involved kidney becomes clearer than that of the other uninvolved kidney (Figure 2).

### 5.2. Color Doppler US

Color Doppler US shows mainly blue or red signals in the normal renal artery. The brightness of these signals is increased in the renal artery (Table 4). The speed of blood flow increases as RAS becomes severe (Figure 3). These Doppler signals show mixed bright red and bright blue colors in the poststenotic dilatation because turbulence is created from the back-and-forth high-speed blood flows out of the stenosis by means of colliding with the lumen of renal artery (Figure 2 and Figure 3).

Color Doppler US of the kidneys is not an ideal approach for detecting RAS. Renal perfusion can be normal in early stage RAS (Figure 4) and decreases in intermediate or late-stage RAS. Renal perfusion is an indirect finding suggesting RAS. Accordingly, it is not adequately sensitive for detecting early stage RAS.

### 5.3. Spectral Doppler US

Spectral Doppler US quantitatively measures the velocity of blood flow in the stenotic renal artery (Table 4). The peak systolic velocity (PSV) within the stenotic renal artery is frequently more than 180–200 cm/s [4,22,23,24,25,48] (Figure 2 and Figure 3). If PSV is more than 180 cm/s, the sensitivity and specificity for RAS range from 85–97% and 72–98%, respectively [24,25,28,31]. In case of unilateral RAS, PSV is significantly different between the two renal arteries. At this point, two simple concepts should be kept in mind: first, the normal PSV values in normal renal arteries without stenosis (about 70–100 cm/s), and, second, angle correction is essential to obtain reproducible and accurate measurements of PSVs. If the PSV of the renal artery is ≥3.5-fold that of the aorta (renal-to-aorta ratio (RAR)), it can suggest RAS [26,27,29,49] (Figure 4). The PSV RAR is another good indicator for identifying RAS. If RAR is 3.5 or greater, the sensitivity and specificity ranges are 91–92% and 71–95%, respectively [26,27,29]. It is important to determine where PSV is measured within the aorta because it differs by region-of-interest location. First, radiologists or sonographers should find the origin of the superior mesenteric artery (SMA) during the sagittal US scan; then, the PSV region of interest should be 1–2 cm below the SMA because the renal arteries are located below the origin of the SMA.

Various kinds of arterial spectra and PSVs can be detected in the post-stenotic dilatation (Figure 2). The Doppler US features results from back-and-forth turbulent flows in the poststenotic dilation [30]. When a renal artery is occluded, the PSV is reduced or absent [23,49]. This severe RAS may result in an almost nonfunctioning kidney, in which the renal cortex becomes very thin. For this reason, PSV is not high because of the severely reduced demand for renal perfusion.

The frequency shift from the stenotic artery is not easy to detect with spectral Doppler US as it tends to be small (Figure 2 and Figure 3). There are some technical tips in assessing RAS: First, the critical angle between the renal artery flow and US from transducer should be observed when a good spectrum of arterial flow is not obtained. The optimal critical angle should be kept at 30–60° for detecting a good frequency shift [48]. Second, the sample volume size should be higher than that of the RAS to oversample frequency shifts. The renal artery spectrum can be identified by means of evaluating the flow direction and spectral pattern from among the various spectra given by arteries and veins. Third, the PSV of the renal artery should be measured in the poststenotic dilatation when a good spectrum cannot be obtained in the stenosis (Figure 2 and Figure 3). If the PSV is more than 180–200 cm/s, it will be higher in the stenotic artery. If RAS is severe, the frequency shifts are too weak to directly detect from the stenotic artery [23,49].

Spectral Doppler US in the kidneys has been performed for indirectly detecting RAS. Several quantitative criteria have been reported in directly detecting RAS, and they include loss of early systolic peak, acceleration index lower than 3 m/s^2^, acceleration time < 0.07 s, and different resistive (>5%) or pulsatile indices (>0.12) between the bilateral kidneys [50,51]. Unlike in renal artery US, there is a poor correlation between renal arteriography and renal Doppler US regarding the detection and degree of RAS. Inter- and intraobserver agreement using these criteria are not high [51,52]. As the renal artery narrows, the systolic peak becomes delayed and weak. Accordingly, the intrarenal spectrum shows pulsus tardus and parvus. This typical finding of RAS is specific, but not sensitive, for detecting RAS. Early stage RAS may not show pulsus tardus and parvus patterns in the kidneys (Figure 3). Several investigations have reported that angioplasty or stenting may not be effective in improving hypertension or renal function in RAS patients who have a pulsus tardus and parvus spectrum [53,54,55].

### 5.4. Contrast-Enhanced US

Compared with CT or MRI contrast material, US contrast material does not harm patients with poor renal function because it does not influence renal function or induce nephrogenic systemic fibrosis. US contrast material is composed of microbubbles, which are destroyed with US and excreted from the pulmonary circulation [56]. Therefore, it does not deteriorate renal function in patients with chronic kidney disease. Initially, this US technique was frequently used for differentiating renal masses [57,58,59,60,61]. The use of US contrast is expanding to assess renal microcirculation for the detection of chronic ischemia [62,63,64,65]. RAS induces decreased blood flow to the renal cortex, which is more susceptible to ischemia than the renal medulla. Moreover, high-frame-rate, contrast-enhanced US can show changes in perfusion parameters, and the shape of the time–intensity curve is useful for assessing cortical perfusion after angio-intervention [32]. Finally, cortex thinning occurs after RAS is persistent. When it is intravenously injected, the renal artery can be imaged as if it were shown on DSA. Contrast-enhanced US can be called “US angiography” if it is used for vascular imaging. Accordingly, the renal artery can be hemodynamically assessed with contrast-enhanced US.

Contrast-enhanced US can be used to directly depict stenosis in the renal artery and poststenotic dilatation [23,66] (Figure 5). In addition, the US signal of an involved kidney is lower than that of the other kidney in the case of unilateral RAS [67]. The renal artery is continuously depicted when the mechanical index is low. Additionally, if a flash mode is selected during the contrast-enhanced US scan, renal artery imaging can be restarted because the in-plane US contrast material is almost all destroyed over a short period. Contrast-enhanced US can depict the renal artery only in the beginning. Renal veins and many adjacent vessels are immediately visualized following renal circulation, and, as such, may prevent precise assessment of RAS. This flash mode has the limitation where the US signal of the renal artery is weaker than that of the renal artery in the beginning of contrast-enhanced US.

## 6. Diagnostic Steps for RAS

Radiologists or sonographers should be familiar with the following steps for detecting RAS: First, they should identify the left renal vein as the first step in detecting bilateral renal arteries with US (Figure 6). If the right or left renal artery is detected on the US, it should be carefully evaluated with gray-scale US. When RAS is detected with gray-scale US, color and spectral Doppler US also should be performed to depict the imaging features of RAS. However, even though RAS is not identified with gray-scale US, renal arteries must be assessed with color Doppler US to detect RAS. RAS may be staged earlier when it is negative on gray-scale US, but positive on color or spectral Doppler US. Angioplasty or stenting is more effective in gray-scale US-negative RAS than in gray-scale US-positive RAS.

Next, the size, echogenicity, and perfusion of kidneys are assessed with renal US when RAS is detected, and assessed with renal artery US. When the size or echogenicity of the involved kidney looks normal, RAS may be in the early stage. Angioplasty or stenting is more effective in this case of RAS. In contrast, if the involved kidney is small or hyperechoic, intervention is not effective because these findings suggest a more advanced stage of RAS.

When renal arteries are not visible on US, the patient’s position needs to be changed to move bowel gas. Even if renal arteries are not still visible after position changes, radiologists or sonographers should perform renal US to assess kidney size, echogenicity, perfusion, or intrarenal spectrum. We have never experienced RAS that is negative on renal artery US but positive on renal US. Because learning to conduct renal artery US has a steep learning curve, beginners may encounter the clinical situation of not finding the renal arteries. Therefore, basic knowledge of renal artery US must be learned first, and practitioners should expect to practice the application of that knowledge in the clinical environment.

## 7. Conclusions

CT, MRI, and DSA using contrast material are rarely recommended for patients who have poor renal function due to RAS. Hence, renal artery US is useful as a primary examination for RAS scanning. Radiologists or sonographers can assess renal arteries once they find the left renal vein behind which the right and left arteries travel. Therefore, RAS can be determined with various imaging features on gray-scale, color Doppler, spectral Doppler, and contrast-enhanced US. The direct assessment using renal artery US is more sensitive to detecting RAS compared with assessment by renal US. In addition, the ongoing development of US scanners has and will provide better hemodynamic information on RAS in patients who cannot undergo contrast-enhanced CT or MR angiography.

## Figures and Tables

**Figure 1 jcm-11-03961-f001:**
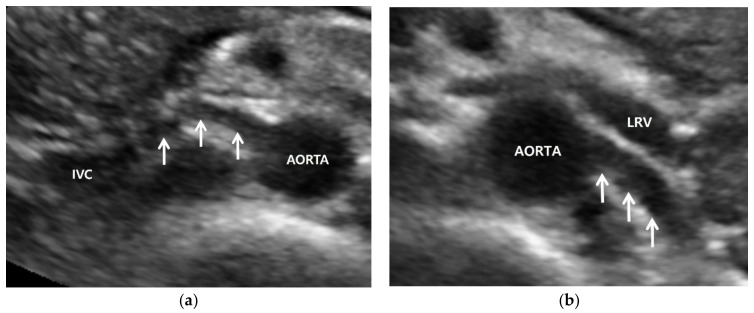
Normal anatomy of renal artery. (**a**) Gray-scale US axial image shows right renal artery (arrows) arising 10 o’clock from the aorta. It shows a short segmental angulation behind the inferior vena cava (IVC); (**b**) gray-scale US axial image shows left renal artery arising 4 o’clock from the aorta. It is located below the left renal vein (LRV) and is traveling away from the transducer without angulation.

**Figure 2 jcm-11-03961-f002:**
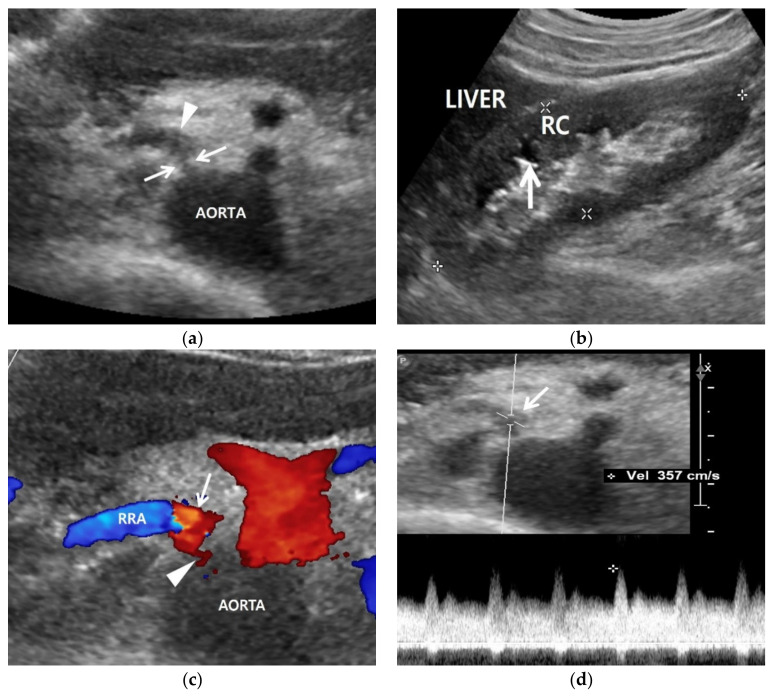
Renal artery and renal US examinations of a 50-year-old man. (**a**) Gray-scale US axial image shows focal stenosis (arrows) in the proximal right renal artery (RRA) and poststenotic dilatation (arrowhead). His RAS was incidentally detected in the routine check-up because his clinical or laboratory findings were unclear. (**b**) Gray-scale US sagittal image that shows a small (9 cm) right kidney in which the cortex (RC) is more hyperechoic compared with the liver parenchyma. Arrows indicate clear cortico-medullary differentiation in the right kidney. (**c**) Color Doppler US shows a focal stenosis (arrowhead) in the proximal renal artery. Bright red and blue signals are seen in the poststenotic dilatation (arrow). (**d**) Spectral Doppler US shows a high PSV (357 cm/s) in the poststenotic dilatation (arrow). However, a low PSV (108 cm/s) was measured in the stenotic artery because the frequency shift from the RAS was not sufficient.

**Figure 3 jcm-11-03961-f003:**
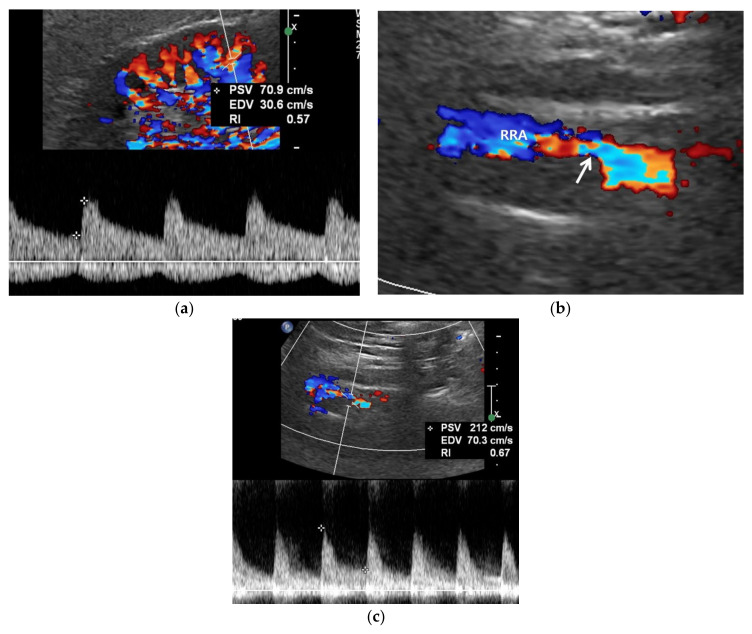
RAS-positive renal artery US in a 20-year-old man with negative renal US. (**a**) Spectral Doppler US does not show pulsus tardus and parvus pattern in the right kidney even though the size (11 cm) and echogenicity appear normal. (**b**) Color Doppler US shows a focal stenosis (arrow) in the proximal right renal artery (RRA), suggesting RAS. (**c**) Spectral Doppler US shows a high peak systolic velocity (PSV) (212 cm/s) in the stenotic right renal artery.

**Figure 4 jcm-11-03961-f004:**
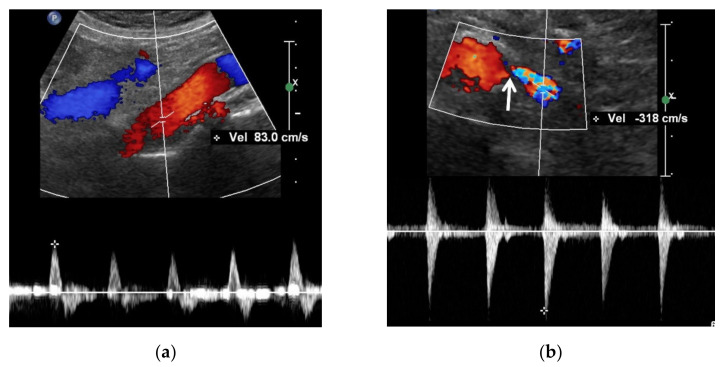
A high reno-aortic PSV ratio in a 78-year-old man. (**a**) Spectral Doppler US shows that a PSV is measured 83 cm/s 1–2 cm below the origin of superior mesenteric artery. (**b**) Spectral Doppler US shows that a PSV is measured 318 cm/s in the poststenotic area, showing turbulence flow. The frequency shift in the proximal left renal artery (arrow) is not sufficient to precisely quantify. The high-PSV RAR is more than 3.8 (318/83) because the PSV in the RAS should be higher than that in the poststenotic area.

**Figure 5 jcm-11-03961-f005:**
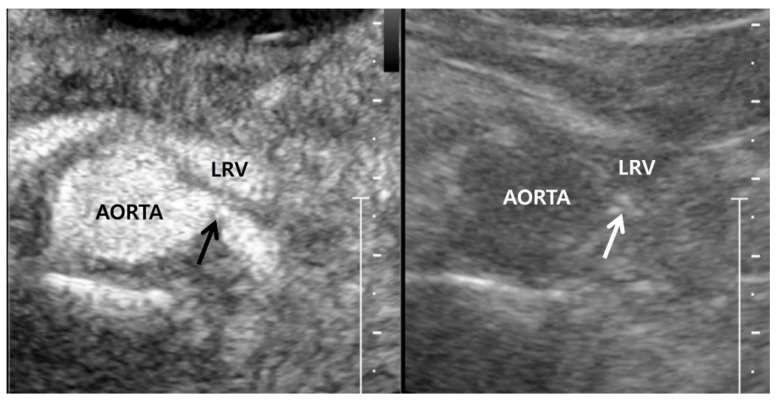
Contrast-enhanced US in a 60-year-old man. Contrast-enhanced US axial image (**left side**), which was obtained 20–30 s after Sonovue (Bracco, Milan, Italy) was intravenously injected, shows a focal stenosis in the proximal left renal artery. Gray-scale US axial image (**right side**), which corresponds to the contrast-enhanced US axial image, shows calcifications (white arrow) in the stenotic wall of the proximal left renal artery. LRV, left renal vein.

**Figure 6 jcm-11-03961-f006:**
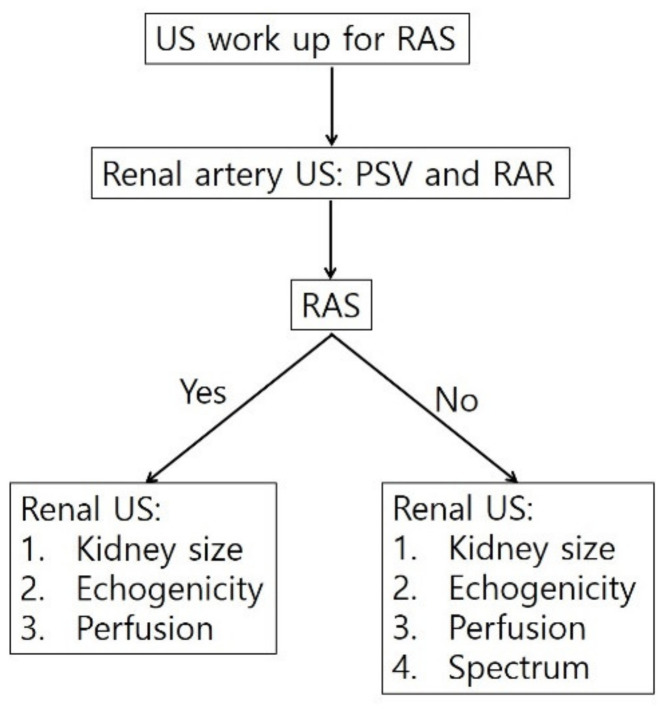
Diagnostic steps for detecting RAS. The flow diagram shows that assessing the renal artery is the first step for diagnosing RAS with gray-scale US, color or spectral Doppler US, or contrast-enhanced US. Next, renal US should be performed to assess kidney size and cortical echogenicity. When RAS is indeterminate on renal artery US, kidneys should be assessed with gray-scale US, color or spectral Doppler US, or contrast-enhanced US. RAS, renal artery stenosis; PSV, peak systolic velocity; RAR, reno-aortic ratio.

**Table 1 jcm-11-03961-t001:** Main etiologies of RAS: differences regarding demographics and treatment.

Demographics	Atherosclerosis	Fibromuscular Dysplasia
Incidence	90%	10%
Sex ratio	No predominance	Women predominant
Common age	Older	Younger
Frequent location	Proximal one-third	Mid-to-distal two-thirds
Main imaging features	Single RAS	Multifocal RAS (beaded)
First treatment option	Angioplasty or stenting	Angioplasty

Note: RAS, renal artery stenosis.

**Table 2 jcm-11-03961-t002:** Renal artery US versus renal US in detecting for RAS.

US Techniques and Accuracy	Renal Artery US	Renal US
Imaging techniques	More difficult	Less difficult
Scan time	Longer	Shorter
Breath hold	Unnecessary	Necessary
Bowel artifact	Frequent	Infrequent
Diagnostic performance	Higher	Lower

Note: RAS, renal artery stenosis.

**Table 3 jcm-11-03961-t003:** Imaging features of renal artery stenosis on gray-scale US.

Imaging Features	Renal Artery US	Renal US
Direct signs	Focal stenosis or occlusion	NA
	Hyperechoic thick wall	NA
Indirect signs	Poststenotic dilatation	Unilateral small kidney
	Calcification	Hyperechoic thin cortex
		Clear CM differentiation

Note: NA, not applicable; CM, cortico-medullary.

**Table 4 jcm-11-03961-t004:** Imaging features of renal artery stenosis on Doppler US.

Doppler US	Renal Artery US	Renal US
Color Doppler US	Bright blue or red	Normal/weak/no perfusion
	Turbulent signal	
Spectral Doppler US	PSV (>180–200 cm/s)	Pulsus tardus and parvus
	Reno-aortic PSV ratio (>3.5)	Delayed acceleration
	Turbulent spectrum	Low resistive index

Note: PSV, peak systolic velocity.

## Data Availability

Not applicable.

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
