# Peer review of "Gray-Scale, Color Doppler, Spectral Doppler, and Contrast-Enhanced Renal Artery Ultrasound: Imaging Techniques and Features"

_jcm, 2022, doi:10.3390/jcm11143961_

Round 1

Reviewer 1 Report

Thank you for submitting this paper on imaging RAS with ultrasound. The majority of the manuscript is a well written technical review that summarizes much of the previous history. I was disappointed not to see more details on how recent imaging technology, such as high-frame rate contrast ultrasound and other features could improve the use of US in RAS further. Additionally, I have the following suggestions:

Introduction:

Page 1, line 26 - please expand your comment on why CT and MR are preferred. Radiologist familiarity is not the only reason. There are many other limitations with Ultrasound. Additionally, you do not comment on the advantage that digital subtraction has with the ability to progress to immediate stenting.

Page 1, line 39 - please expand on these original US limitations and consider how modern high-end cart based US platforms address these. This is important for context as you then list the technology (higher resolution).

I feel a general paragraph on the operator dependence of US, the necessary education and learning curve for RAS scanning is needed before your paragraph on page 2, lines 50-52.

Clinical Aspects of RAS:

Page 2, line 55 - Reference 30 is twenty-two years old. There are more recent manuscripts on RAS epidemiology.

I feel this section could be improved with a paragraph or two on the treatment options for RAS. It is my experience that imaging for RAS is generally restricted to those patients who's hypertension is resistant to medication. Especially important given the results of the CORAL trial.

Renal artery US: Imaging Techniques:

You described limited anatomy and do not consider anatomical variants or accessory & polar renal vessels. These should be described.

The sections on Grey-Scale and Colour Doppler are to the point and well written.

Spectral Doppler US: (line numbers are incorrectly displayed)

Consider re-drafting the first two lines of the second paragraph. Otherwise a technically well written section.

Contrast US section is also well written.

Renal artery US versus renal US:

Sections of these paragraphs should come much earlier in the manuscript. Particularly the comments on sub-segmental renal arteries (see comment about introduction). 

Diagnostic steps for RAS:

Elements of this are discussed earlier in the manuscript and it appears to be repetition. Consider restructuring the manuscript to avoid this repetition.

Conclusion:

Needs a statement at the start on the limitations of CT, MR and DSA. A general statement on how US can address these is needed.

Author Response

Reviewer#1 comments

Thank you for submitting this paper on imaging RAS with ultrasound. The majority of the manuscript is a well written technical review that summarizes much of the previous history. I was disappointed not to see more details on how recent imaging technology, such as high-frame rate contrast ultrasound and other features could improve the use of US in RAS further. Additionally, I have the following suggestions:

Response: Thank you for your comments. I will add the advantage of high-frame rate contrast ultrasound and other features in the section of contrast-enhanced US section.

Introduction:

Page 1, line 26 - please expand your comment on why CT and MR are preferred. Radiologist familiarity is not the only reason. There are many other limitations with Ultrasound. Additionally, you do not comment on the advantage that digital subtraction has with the ability to progress to immediate stenting.

Response: Thank you for your comment. In patients with normal renal function, CT and MRI can be performed because these imaging modalities are objective and can provide functional images. However, renal artery ultrasound is a screening examination for RAS and thus should not be omitted prior to CT and MRI scans. Moreover, CT and MRI need injection of contrast materials for detecting RAS. Therefore, these imaging modalities cannot avoid additional cost and complications even though these are reported to be low.

Real advantage of renal artery ultrasound is that it can be performed in patients with poor renal function. Because symptomatic patients with RAS have decreased renal function, iodine or gadolinium contrast material is not recommended to inject.

I will add comment on the advantage of digital subtraction for detecting RAS, but this imaging modality also needs to use iodine contrast material. For the same reason, it is not recommended to use in symptomatic patients with RAS.

Page 1, line 39 - please expand on these original US limitations and consider how modern high-end cart based US platforms address these. This is important for context as you then list the technology (higher resolution).

I feel a general paragraph on the operator dependence of US, the necessary education and learning curve for RAS scanning is needed before your paragraph on page 2, lines 50-52.

Response: Thank you for your comments. I will add additional US limitations and the advantages on high-end cart based US platforms.

Clinical Aspects of RAS:

Page 2, line 55 - Reference 30 is twenty-two years old. There are more recent manuscripts on RAS epidemiology.

I feel this section could be improved with a paragraph or two on the treatment options for RAS. It is my experience that imaging for RAS is generally restricted to those patients who's hypertension is resistant to medication. Especially important given the results of the CORAL trial.

Response: Thank you for your comment. I will add one or two paragraphs on the treatment options and the CORAL trial is added as a reference.

Renal artery US: Imaging Techniques:

You described limited anatomy and do not consider anatomical variants or accessory & polar renal vessels. These should be described.

Response: Thank you for your comment. I will add the statement on anatomical variant or accessory and polar arteries.

The sections on Grey-Scale and Colour Doppler are to the point and well written.

Spectral Doppler US: (line numbers are incorrectly displayed)

Consider re-drafting the first two lines of the second paragraph. Otherwise a technically well written section.

Response: Thank you for your comments. I will rephrase them for better understanding.

Contrast US section is also well written.

Renal artery US versus renal US:

Sections of these paragraphs should come much earlier in the manuscript. Particularly the comments on sub-segmental renal arteries (see comment about introduction). 

Response: Thank you for your comments. I will move this section behind the clinical aspect of RAS.

Diagnostic steps for RAS:

Elements of this are discussed earlier in the manuscript and it appears to be repetition. Consider restructuring the manuscript to avoid this repetition.

Response: Thank you for your comment. I will minimize the repetition.

Conclusion:

Needs a statement at the start on the limitations of CT, MR and DSA. A general statement on how US can address these is needed.

Response: Thank you for your comments. I will add it at the beginning.

Reviewer 2 Report

Imaging tests commonly done to diagnose renal artery stenosis include:

  • Doppler ultrasound. High-frequency sound waves help your doctor see the arteries and kidneys and check their function. This procedure also helps your doctor find blockages in the blood vessels and measure their severity.
  • CT scan. During a CT scan, an X-ray machine linked to a computer creates a detailed image that shows cross-sectional images of the renal arteries. You may receive a dye injection to show blood flow.
  • Magnetic resonance angiography (MRA). MRA uses radio waves and strong magnetic fields to produce detailed 3D images of the renal arteries and kidneys. A dye injection into the arteries outlines blood vessels during imaging.
  • Renal arteriography. This special type of X-ray exam helps your doctor find the blockage in the renal arteries and sometimes open the narrowed part with a balloon or stent. Before an X-ray is taken, your doctor injects a dye into the renal arteries through a long, thin tube (catheter) to outline the arteries and show blood flow more clearly. This test is mainly done if it's also likely that you need a small tube (stent) placed in your blood vessel to widen it.
  • Ultrasound

    Ultrasound, although most freely available, cheap and often used first line, is relatively operator-dependent and may prove time-consuming.

    • increased peak systolic velocity (PSV): some advocate 180 cm/s 4
    • increased renal-interlobar ratio (RIR), i.e. PSVrenal artery (intrastenotic)/PSVinterlobar (distal): some advocate values greater than 5 3
    • increased renal-aortic ratio (RAR), i.e. PSVrenal/PSVaorta: usually taken as >3.5, although some advocate >3 4 or even >2 3
      • ​lower cut off values increase sensitivity but decrease specificity
    • turbulent flow in a post-stenotic area
    • pulsus parvus et tardus waveform (slow-rising) due to stenosis
    • decreased (interlobar) renal arterial resistive index (RI): <0.55 in severe stenosis 10
    • resistive index difference between kidneys >5 % 9
    • intraparenchymal acceleration time >0.07 s
    • acceleration index (AI): lower than 3 m/s2
    CT angiography

    The three-dimensional reconstruction of the renal vascular tree provides a reliable method of visualizing the entire vascular tree. Images are acquired with thin collimation and bolus tracking on the abdominal aorta. Sensitivity and specificity varying between 90 to 99% have been reported 7.  Both the raw data and 3D reconstructions should be viewed. Additionally, supernumerary arteries may be identified.

    MR angiography

    Different imaging methods can be used for renal MRA:

    • time of flight (TOF): whereby the high velocity of the blood jet at the level of stenosis appears as a loss of signal (black)
    • phase contrast technique
    • contrast-enhanced MRA: gadolinium is used as a contrast agent

    Three-dimensional reconstruction technique offers sensitivity and specificity values around 90 to 100% 7. In some cases, renal impairment does not permit the use of contrast, in which case TOF imaging is beneficial.

    Reported sensitivity and specificity for MR angiography is at around > 95% and > 90% for detection of stenoses of 50% or greater in diameter. MR angiography may overestimate moderate stenosis and detection / evaluation of accessory and branch arteries can at times be problematic.

    Nuclear medicine
    ACE inhibitor scintigraphy
    • the affected kidney with renovascular hypertension shows impaired function due to ACE inhibition; based on this principle scintigraphy has been very much useful for diagnosis of renal artery stenosis
    • performed by IV administration of enalapril maleate after 15 minutes
    • sequential images and scintigraphic curves are plotted for the renal cortex and pelvis; renal uptake is measured for every 1-2 min interval after administering the IV injection
    • typical isotopes used are Tc-99m MAG3, Tc-99m DTPA or I-123 ortho-iodohippurate 6
    • interpreted as either low, intermediate or high probability

  • References should be enriched
  • Li J, Wang L, Jiang Y et al. Evaluation of Renal Artery Stenosis with Velocity Parameters of Doppler Sonography. J Ultrasound Med. 2006;25(6):735-42; quiz 743-4.

Author Response

Reviewer#2 comments

Imaging tests commonly done to diagnose renal artery stenosis include:

  • Doppler ultrasound. High-frequency sound waves help your doctor see the arteries and kidneys and check their function. This procedure also helps your doctor find blockages in the blood vessels and measure their severity.
  • CT scan. During a CT scan, an X-ray machine linked to a computer creates a detailed image that shows cross-sectional images of the renal arteries. You may receive a dye injection to show blood flow.
  • Magnetic resonance angiography (MRA). MRA uses radio waves and strong magnetic fields to produce detailed 3D images of the renal arteries and kidneys. A dye injection into the arteries outlines blood vessels during imaging.
  • Renal arteriography. This special type of X-ray exam helps your doctor find the blockage in the renal arteries and sometimes open the narrowed part with a balloon or stent. Before an X-ray is taken, your doctor injects a dye into the renal arteries through a long, thin tube (catheter) to outline the arteries and show blood flow more clearly. This test is mainly done if it's also likely that you need a small tube (stent) placed in your blood vessel to widen it.
  • Ultrasound

Ultrasound, although most freely available, cheap and often used first line, is relatively operator-dependent and may prove time-consuming.

    • increased peak systolic velocity (PSV): some advocate 180 cm/s 4
    • increased renal-interlobar ratio (RIR), i.e. PSVrenal artery (intrastenotic)/PSVinterlobar (distal): some advocate values greater than 5 3
    • increased renal-aortic ratio (RAR), i.e. PSVrenal/PSVaorta: usually taken as >3.5, although some advocate >3 4 or even >2 3
      • ​lower cut off values increase sensitivity but decrease specificity
    • turbulent flow in a post-stenotic area
    • pulsus parvus et tardus waveform (slow-rising) due to stenosis
    • decreased (interlobar) renal arterial resistive index (RI): <0.55 in severe stenosis 10
    • resistive index difference between kidneys >5 % 9
    • intraparenchymal acceleration time >0.07 s
    • acceleration index (AI): lower than 3 m/s2

CT angiography

The three-dimensional reconstruction of the renal vascular tree provides a reliable method of visualizing the entire vascular tree. Images are acquired with thin collimation and bolus tracking on the abdominal aorta. Sensitivity and specificity varying between 90 to 99% have been reported 7.  Both the raw data and 3D reconstructions should be viewed. Additionally, supernumerary arteries may be identified.

MR angiography

Different imaging methods can be used for renal MRA:

    • time of flight (TOF): whereby the high velocity of the blood jet at the level of stenosis appears as a loss of signal (black)
    • phase contrast technique
    • contrast-enhanced MRA: gadolinium is used as a contrast agent

Three-dimensional reconstruction technique offers sensitivity and specificity values around 90 to 100% 7. In some cases, renal impairment does not permit the use of contrast, in which case TOF imaging is beneficial.

Reported sensitivity and specificity for MR angiography is at around > 95% and > 90% for detection of stenoses of 50% or greater in diameter. MR angiography may overestimate moderate stenosis and detection / evaluation of accessory and branch arteries can at times be problematic.

Nuclear medicine

ACE inhibitor scintigraphy

    • the affected kidney with renovascular hypertension shows impaired function due to ACE inhibition; based on this principle scintigraphy has been very much useful for diagnosis of renal artery stenosis
    • performed by IV administration of enalapril maleate after 15 minutes
    • sequential images and scintigraphic curves are plotted for the renal cortex and pelvis; renal uptake is measured for every 1-2 min interval after administering the IV injection
    • typical isotopes used are Tc-99m MAG3, Tc-99m DTPA or I-123 ortho-iodohippurate 6
    • interpreted as either low, intermediate or high probability

  • References should be enriched
  • Li J, Wang L, Jiang Y et al. Evaluation of Renal Artery Stenosis with Velocity Parameters of Doppler Sonography. J Ultrasound Med. 2006;25(6):735-42; quiz 743-4.

Response: Thank you for your comments. I will add some references as you recommend. My review article is focused on symptomatic patients with RAS. They mostly have reduced renal function so that they cannot undergo contrast enhanced CT, MRI, and conventional arteriography because of contrast-induced complications. Therefore, renal artery US is a primary examination to screen RAS. These statement will be added in the text.

Round 2

Reviewer 1 Report

Corrections have been handled well. Only very minor grammar tweaks are needed but this can be done at style editing.

Author Response

Authors’ response to Reviewer#1 comments:

Corrections have been handled well. Only very minor grammar tweaks are needed but this can be done at style editing.

Response: Thank you for your comment. Actually, original manuscript has been edited by professional editing service. It was revised by myself before re-submission because it was supposed to submit within 10 days after I received a decision letter. Therefore, there are minor grammar errors because I am not a native speaker. This revision also should be submitted within 5 days, so that I cannot professional editing service. As a result, I will try my best in correcting grammatical or spelling errors. I hope that it will become perfect at style editing.